# Study of Humoral Responses against *Lomentospora*/*Scedosporium* spp. and *Aspergillus fumigatus* to Identify *L. prolificans* Antigens of Interest for Diagnosis and Treatment

**DOI:** 10.3390/vaccines7040212

**Published:** 2019-12-10

**Authors:** Idoia Buldain, Aize Pellon, Beñat Zaldibar, Aitziber Antoran, Leire Martin-Souto, Leire Aparicio-Fernandez, Maialen Areitio, Emilio Mayayo, Aitor Rementeria, Fernando L. Hernando, Andoni Ramirez-Garcia

**Affiliations:** 1Fungal and Bacterial Biomics Research Group, Department of Immunology, Microbiology and Parasitology, Faculty of Science and Technology, University of the Basque Country (UPV/EHU), Leioa 48940, Spain; idoia.buldain@ehu.eus (I.B.); aize.pellon@ehu.eus (A.P.); aitziber.antoran@ehu.eus (A.A.); leire.martin@ehu.eus (L.M.-S.); leire.aparicio@ehu.eus (L.A.-F.); mareitio004@ikasle.ehu.eus (M.A.); andoni.ramirez@ehu.eus (A.R.-G.); 2CBET Research Group, Department of Zoology and Animal Cell Biology, Faculty of Science and Technology, Research Centre for Experimental Marine Biology and Biotechnology PIE, University of the Basque Country (UPV/EHU), Leioa 48940, Spain; benat.zaldibar@ehu.eus; 3Pathology Unit, Medicine and Health Science Faculty, University of Rovira i Virgili, Reus 43201, Spain; emilio.mayayo@urv.cat

**Keywords:** electrophoresis, mass spectrometry, proteomic, omic, cross-reactivity, 2DE, western blot

## Abstract

The high mortality rates of *Lomentospora prolificans* infections are due, above all, to the tendency of the fungus to infect weakened hosts, late diagnosis and a lack of effective therapeutic treatments. To identify proteins of significance for diagnosis, therapy or prophylaxis, immunoproteomics-based studies are especially important. Consequently, in this study murine disseminated infections were carried out using *L. prolificans*, *Scedosporium aurantiacum*, *Scedosporium boydii* and *Aspergillus fumigatus*, and their sera used to identify the most immunoreactive proteins of *L. prolificans* total extract and secreted proteins. The results showed that *L. prolificans* was the most virulent species and its infections were characterized by a high fungal load in several organs, including the brain. The proteomics study showed a high cross-reactivity between *Scedosporium/Lomentospora* species, but not with *A. fumigatus*. Among the antigens identified were, proteasomal ubiquitin receptor, carboxypeptidase, Vps28, HAD-like hydrolase, GH16, cerato-platanin and a protein of unknown function that showed no or low homology with humans. Finally, Hsp70 deserves a special mention as it was the main antigen recognized by *Scedosporium/Lomentospora* species in both secretome and total extract. In conclusion, this study identifies antigens of *L. prolificans* that can be considered as potential candidates for use in diagnosis and as therapeutic targets and the production of vaccines.

## 1. Introduction

*Lomentospora prolificans,* formerly known as *Scedosporium prolificans*, is an emerging pathogen with an outstanding ability to disseminate through the bloodstream and cause disseminated infections in severely immunosuppressed patients, particularly those with hematological malignancies. Despite the relatively low prevalence of these infections, they are an important concern for clinicians owing to their negative prognosis, with mortality rates of up to 87.5% [1]. This fact is associated with the tendency of the fungus to infect weakened hosts, the lack of rapid diagnosis methods and its intrinsic resistance to virtually all currently available antifungals. It is typically described as being resistant to echinocandins, pyrimidines, allylamines, polyenes and to a lesser extent, azoles. Currently, European guidelines recommend voriconazole (VRZ) as the treatment of first choice. However, the mean minimal inhibitory concentration of this drug necessary to inhibit 50% of isolates (MIC_50_ = 4 µg/mL) is rarely obtained freely in patients, which is consistent with the poor response to treatment with VRZ in patients infected with *L. prolificans* [2,3].

One of the most promising therapeutic strategies is the combination of different antifungals, which according to the scarce data obtained up to this moment in time, has demonstrated greater efficacy in vitro and in vivo than monotherapy [4]. Of special interest is the synergistic interaction between terbinafine and triazoles (VRZ, itraconazole, or miconazole), which obtain minimal inhibitory concentrations (MICs) achievable in patient serum [3]. However, the use of this combination has brought variable results [5,6,7,8,9]. On the other hand, in recent years, new drugs have been developed with notable in vitro activity against *L. prolificans*, for example Trichostatin A, F901318, E1210 and N-chlorotaurine [4], all of which require further studies to evaluate their efficacy *in vivo*.

Over the last couple of decades great efforts have been made in the search for alternatives to chemotherapy, through the development of monoclonal antibodies or prophylactic vaccines. Although these studies have not so far focused on *L. prolificans*, very promising results have been obtained with other important fungal pathogens such as *Coccidioides immitis, Cryptococcus neoformans* and *Candida albicans* [10]. The development of these alternative therapies requires the identification of new fungal targets and immunoproteomic studies could well be a suitable tool to achieve this objective. Among them, the comparative studies between different microorganisms could be especially significant as they would allow the identification of both species-specific molecules and pan-fungal targets.

Therefore, this study developed a disseminated murine infection model to compare the virulence of *L. prolificans* with those of *Scedosporium boydii*, *Scedosporium aurantiacum* and *Aspergillus fumigatus*. Moreover, to find new candidates for study as therapeutic and diagnostic targets, the immunodominant antigens associated with non-infective and infective doses of *L. prolificans*, as well as the species-specific and those shared by more than one species were detected and identified using immunoproteomics.

## 2. Materials and Methods

### 2.1. Microorganisms and Culture Conditions

In this study the strains of *Lomentospora prolificans* CECT 20842, *Scedosporium boydii* CECT 21169, *Scedosporium aurantiacum* CBS 116910 and *Aspergillus fumigatus* Af293 were used.

All the strains were cryopreserved at −80 °C and cultured onto Potato Dextrose Agar (PDA) (Pronadisa, Madrid, Spain) at 37 °C for 7 days before use. To obtain *L. prolificans*, *S. boydii* and *S. aurantiacum* conidia, the plates were washed in saline buffer solution (0.9% NaCl) in duplicate, then the suspension was filtered through a gauze and centrifuged. The concentration of conidia was adjusted using a hemocytometer to inoculate 5 × 10^5^ conidia/mL in Potato Dextrose Broth (PDB, Pronadisa) and incubated at 37 °C for 7 days. Finally, conidia were collected by filtration through a gauze and then centrifuged (at 11,400× *g*, 5 min and 4 °C). Conidia of *A. fumigatus* were collected from PDA tubes grown at 28 °C for 7 days using saline solution-Tween 20 (0.9% NaCl, 0.02% Tween 20) (ss-tween 20) and washed twice by centrifugation.

### 2.2. Models of Murine Disseminated Infection

Eight-week old Swiss female mice were used, these were bred and maintained at the SGIker Animal Facility of the Animal Experimentation Ethical Committee from the University of the Basque Country (UPV/EHU). Animals were maintained with water and food ad libitum in filter-aerated sterile cages. All the procedures carried out in the assay were approved by the Animal Experimentation Ethical Committee from the University of the Basque Country (UPV/EHU) (M20/2016/235, M20/2016/323).

All infections were made by intravenous injection in the tail vein. To do this, conidia were suspended in ss-Tween 20 with 0.2 mL/animal being administered.

For the development of a murine model of *L. prolificans* disseminated infection, twenty-four mice were divided into five groups, four groups were administered with 10^2^, 10^3^, 10^4^ or 10^5^ conidia/animal and a control group received ss-Tween 20. All groups contained four mice, except the group with the highest dose which, because of the increased mortality detected, required eight individuals.

To perform the comparative studies of *L. prolificans*, *S. boydii*, *S. aurantiacum* and *A. fumigatus* intravenous murine infections, a total of 48 mice were intravenously injected with 0.2 mL of ss-Tween 20 (control group) or the indicated dose of fungal conidia. Mice infected with *L. prolificans*, *S. boydii* or *S. aurantiacum* received 10^5^ conidia/animal. In the case of *A. fumigatus*, three different doses were administered: 10^5^, 10^6^ and 5 × 10^6^ conidia/animal. Groups contained six mice, except in the case of those infected with *L. prolificans* and *A. fumigatus* 5 × 10^6^ conidia/animal, which contained twelve mice/group, owning to the high mortality rate observed.

### 2.3. Study of the Infection Process by CFU Counting and Histology

At the end point or 28 days after the inoculation, the animals were sacrificed to extract total blood as well as the following organs: kidneys, lungs, spleen, liver and brain. Blood samples were coagulated, centrifuged (Microvette, Sarstedt, Nümbrecht, Germany) and stored at −80 °C until needed. Organs were divided into two halves, one for fungal load determination by counting the Colony Forming Units (CFU), and the other for the histological study.

To evaluate the fungal load, organs were weighed and then mechanically homogenized in 1 mL ss-Tween 20. Finally, 0.1 mL from the diluted homogenate was inoculated by extension on PDA plates containing 10 μg/mL chloramphenicol (Sigma-Aldrich, St Louis, MO, USA) and 25 μg/mL gentamicin (Sigma-Aldrich) in duplicate. Plates were incubated at 37 °C and the CFU counted after 2–3 days. To perform the histological study, the organs of all the animals were fixed in 10% formalin and immersed in paraffin. Then, at least five different cuts, four micrometers wide, were stained with hematoxylin-eosin and Grocott’s methenamine silver.

### 2.4. Obtaining Total Protein Extract and Secretome of L. prolificans

To collect the total protein extract, 5 × 10^5^ conidia/mL were inoculated into 150 mL PDB at 37 °C and 120 rpm for 24 h. Then, the culture was centrifuged at 11,400× *g* and 4 °C for 5 min, and washed twice with phosphate buffer saline (PBS). The cell pellets were resuspended in PBS plus 1% (*v*/*v*) 2-mercaptoethanol and 1% (*v*/*v*) ampholites pH 3–10 (GE Healthcare, Freiburg, Germany). Cell disruption was performed following the protocol described in Buldain et al. (2016) [11].

Turning to the method used to extract the *L. prolificans* secretome, different culture conditions and methodologies were tested to select the most adequate. On the one hand, 10^6^ conidia/mL were inoculated into 300 mL of Sabouraud Dextrose Broth (SDB) (Panreac, Barcelona, Spain), PDB, and yeast extract and glucose medium (0.5% (*p*/*v*) yeast extract, 2% (*p*/*v*) glucose; filtered through 5 kDa pore). Conidia were incubated at 37 °C and 120 rpm, and cultures were collected at 3, 7, 14 and 21 days. Cultures were filtered through filter paper, sterilized using 0.22 μm filters (Merck Millipore, Cork, Ireland) and concentrated with Sartorius ultrafiltration equipment (Göttingen, Germany). It was then dialyzed (Orange Scientific, Belgium) and centrifuged (11,400× *g*, 30 min) to discard the insoluble fraction.

In parallel, the methodology employed by da Silva et al. (2012) [12] was also carried out, but with small modifications. Briefly, 10^6^ conidia/mL were inoculated into 300 mL PDB at 37 °C and 120 rpm for 24 h. The fungus was collected and cultured in PBS plus 2% glucose at 37 °C and 120 rpm for 20 h. Finally, the supernatant was collected and sterilized using 0.22 μm filters.

The collection of all the extracts was made in triplicate. Protein concentration was quantified using Pierce 660 nm Protein Assay Reagent (Thermo Fisher Scientific, Rockford, IL, USA) and the quality of the extract checked by SDS-PAGE, in 12.5% acrylamide gels. The gels were stained with Coomassie Brilliant Blue (CBB) as previously described [13] and digitalized using ImageScanner III (GE Healthcare).

### 2.5. Verification of Cellular Integrity and Absence of Cytoplasmic Proteins

The cellular integrity of the fungus was studied in order to confirm the absence of cytoplasmic proteins in the secretome. To achieve this, fungal samples were observed under optical microscopy, and an enzymatic assay was performed in the supernatant using the Lactate Dehydrogenase enzyme (LDH). For this, the substrate (10 μL 1M pyruvate, 770 μL 100 mM Tris-HCl (pH 7.1)) was pre-incubated at 30 °C for 5 min. Then, 20 μL of 15 mM reduced nicotinamide adenine dinucleotide (NADH) and 200 μL of the samples were added to the incubation medium and absorbance at 340 nm was measured every 60 s for 5 min. Pure LDH was used as positive control.

### 2.6. Protein Detection by Two-Dimensional Electrophoresis

Protein precipitation of total extract was performed as in Buldain et al. (2016) [11]. In the case of the secretome, it was carried out at −20 °C in one volume of acetone plus 20% (*p*/*v*) trichloroacetic acid and 0.07% (*v*/*v*) 2-mercaptoethanol for 1 h. The sample was then centrifuged (11.400× *g*, 45 min, 4 °C), the supernatant discarded and the pellets washed three times with cold acetone, air-dried and resuspended in rehydration buffer (7 M urea, 2 M thiourea, 20 mM Tris, 4% (*p*/*v*) CHAPS, 0.5% (*v*/*v*) ampholyte, 20 mM DTT, 0.002% (*p*/*v*) bromophenol blue). Samples were sonicated using 2 pulses at an amplitude of 40 for 2 min, and stored at −20 °C until further use.

Afterwards, the isoelectric focusing (IEF) was carried out using 18 cm long Immobiline DryStrip gels (pH 3–10, GE Healthcare) and ReadyStripTM IPG Strips pH 3–6 (Bio-Rad), loaded with 400 µg of protein. All the two-dimensional electrophoresis (2DE) were made in triplicate and only the most informative gels of the two pH ranges used are shown in the results. The protocol described in Pellon et al. (2016) [14] was followed for the IEF and 2DE of the total extract, using 12.5% gels. For the secretome, an initial step of 150 V at 300 Vhr in the IEF protocol and the use of 13% polyacrylamide gels in the second dimension were the only modifications introduced.

### 2.7. Antigenic Detection

Gels were transferred to Amersham Hybond-P polyvinylidene difluoride (PVDF) membranes (GE Healthcare) at 400 mA for 2 h. In order to confirm the correct transference of the proteins, membranes were stained with Ponceau Red (0.2% (*p*/*v*) Ponceau Red, 1% (*v*/*v*) acetic acid). For antigen detection, membranes were blocked in Tris-Buffered Saline (TBS) plus 5% (*p*/*v*) skimmed milk and 0.1% (*v*/*v*) Tween 20 (TBSM) for 2 h. Then they were incubated overnight with mouse serum diluted in TBSM at 4 °C. After that, membranes were washed four times for 5 min with TBS and then incubated with murine anti-IgG-HPR diluted to 1/100,000 in TBSM. All the incubations were made at room temperature, unless otherwise stated. Immunoreactive proteins were detected using ECL Plus (GE Healthcare) following the manufacturer instructions in the G:BOX Chemi system (Syngene). Western Blot (WB) analysis was carried out using ImageMaster 2D Platinum Software (GE Healthcare).

### 2.8. Identification of Immunoreactive Proteins

The most immunodominant antigens were manually extracted from CBB stained gels and identified by LC-MS/MS in the SGIker services of the UPV/EHU, as described in Buldain et al. (2016) [11]. The search for protein identification was made in the non-redundant database of the NCBI, restricted to fungi, using the online server MASCOT (Matrix Science Ltd., London, UK; http://www.matrixscience.com). When more than one result is obtained at the same spot, the only proteins shown are those with a MASCOT score greater than 60% of the data obtained for the best identified protein from the spot, and with a coverage of ≥5.

### 2.9. Bioinformatic Analysis of L. prolificans Antigens

In order to predict the location, the secretion pathway, and the possible adhesion properties of the antigens, TargetP (http://www.cbs.dtu.dk/services/TargetP/), SecretomeP 2.0 (http://www.cbs.dtu.dk/services/SecretomeP/), and FaaPred (http://bioinfo.icgeb.res.in/faap/) software were used, respectively. Results were considered positive when score was ≥0.5 for TargetP, ≥0.6 for SecretomeP 2.0, or setting a −0.8 threshold for FaaPred. Finally, to study the functionality, family and domains, the Interpro database was used (http://www.ebi.ac.uk/interpro/).

### 2.10. Heat Shock Protein 70 Electroelution

The spots corresponding to the Hsp70 were manually extracted from the Coomassie R-250 (Sigma-Aldrich) stained 2DE gels of the *L. prolificans* total protein extract. Then they were electroeluted with the 422 Electoeluter (Bio-Rad) system, in accordance with manufacturer’s instructions. Finally, a 1DE was made with the electroeluted samples, using 12% gels.

### 2.11. Statistics

The fungal tissue burdens were compared using the two-way ANOVA method followed by multiple comparisons corrected with Dunnett’s test in SPSS (version 17.0 for Windows; Chicago, IL, USA) and plotted using GraphPad Prism version 7 (Graph Prism Software Inc., San Diego, CA, USA).

## 3. Results

### 3.1. Development of a Murine Disseminated Infection Model with L. prolificans

The study of the infection process showed that in the groups of mice inoculated with 10^2^ and 10^3^ conidia/animal a lower number of CFUs were counted, mainly in organs associated with the blood circulation (spleen, kidney, liver) (Figure 1A). In the groups infected with 10^4^ and 10^5^ conidia/animal a higher number of CFUs was collected. Specifically, the differences with the highest dose were statistically significant in all the organs, highlighting the high presence of the fungus in the brain. The kidneys, spleen and brain were the most affected organs with the highest dose, 4.77 ± 1.58, 3.34 ± 0.86 and 3.20 ± 0.86 log CFU/g were recorded, respectively.

Signs of infection were only detected in those animals infected with the highest dose, the most common signs being curved abdomen, ruffled hair, isolation, persistent lethargy, weight loss and neurological alterations. The main neurological sign detected was leaning to one side, which was observed three days after inoculum administration, something that affected 62.5% of the infected individuals (Figure 1B). Weight monitoring showed an inverse relationship with the number of conidia administered (Figure 1C) and mortality was only observed in the group infected with the highest dose, a 50% survival rate being recorded. Due to the results obtained, the highest dose (10^5^ conidia/animal) was established as the infective dose and 10^2^ conidia/animal as a non-infective contact dose.

### 3.2. Comparative Study of Disseminated Murine Infections Produced by L. prolificans, S. boydii, S. aurantiacum and A. fumigatus

Taking into account the proximity between species of the genera *Lomentospora* and *Scedosporium*, 10^5^ conidia/animal was used for all of them. In the case of *A. fumigatus*, 10^5^, 10^6^ and 5 × 10^6^ conidia/animal were studied to select an infective dose, this turned out to be 5 × 10^6^ conidia/animal as it was the only one that showed clear results related to an infection process and mortality.

The comparison of the fungal burden by counting CFUs showed very similar results in both kidneys and spleens from the groups infected with *L. prolificans* and *S. boydii*. Specifically, 5.17 ± 0.74 and 4.48 ± 1.62 log CFU/g were collected in the kidneys for *L. prolificans* and *S. boydii*, respectively, and 3.46 ± 0.90 and 3.74 ± 0.15 log CFU/g in the spleen (Figure 2A). The data from the kidneys were significantly higher than with *S. aurantiacum*. In mice infected with *A. fumigatus*, the kidneys were the most affected organ with a fungal load of 3.44 ± 0.87 log CFU/g. It is also remarkable that while in the group of mice infected with *L. prolificans* the brain was the third organ with the highest fungal load (3.36 ± 0.78 log CFU/g), in the all the other groups the brain was the least infected, these differences being statistically significant.

The study of histological sections showed renal invasion only in groups infected with *Lomentospora* spp. (Figure 3A) and *Scedosporium* spp. (Figure 3B,C). In these groups the pelvis of the kidneys were dilated with a variable amount of inflammatory response by polymorphonuclears and eosinophils, with hyphae in the centre, and a high presence of *L. prolificans* conidia. However, in the groups of mice infected with *A. fumigatus* (Figure 3D) no affectation was observed.

Signs associated with the development of the infection were observed in all groups during the daily monitoring of the animals welfare. To be specific, the same signs associated with animal discomfort and the development of the infection described previously for *L. prolificans* were observed in all of them, but in the group infected with *S. aurantiacum* with a lesser degree of severity. More specifically, leaning to one side was the most widespread of all the signs detected. It was detected in 66.67% of those infected with *L. prolificans*, in 100% of those infected with *S. boydii* or *S. aurantiacum*, and in 66.67% of those infected with *A. fumigatus*. This effect was detected three days after the administration of *L. prolificans*, and 4 days after the administration of *S. boydii, S. aurantiacum* and *A. fumigatus.*

The groups infected with *L. prolificans* and *A. fumigatus* were the groups with the highest mortality rates, 50% (Figure 2B). In both cases the first death was recorded nine days after administration of the inoculum. The groups infected by *S. aurantiacum* and *S. boydii* had a mortality rate of 16.67%, with the first deaths recorded on the 13th and 19th days, respectively. In relation to the weight of mice (Figure 2C), the groups infected with *L. prolificans, S. boydii* and *A. fumigatus* showed the most pronounced weight loss, with the largest decrease observed in the *L. prolificans* group.

### 3.3. Identification of the Immunodominant Antigens of L. prolificans Secretome Recognized by Mice Inoculated with Contact and Infective Dose

In order to obtain the secretome of *L. prolificans* from all the different media and incubation times studied, the cultivation in SDB for 14 days was selected as the number and definition of the 1DE bands were the best (Appendix A). This methodology was, in turn, compared with the protocol described Da Silva et al. (2012) [12] using 2DE, and although the gels had similar patterns the second method had a larger number (373 vs. 253) and a better focusing of spots was achieved (Appendix A). So, after proving that the proteins recovered were not intracellular by observing the cell integrity and assaying the LDH activity (data not shown), the second method was selected for the following experiments.

After this, the reactivity of the humoral response by one-dimensional WB (1DWB) against the secretome and total extract of *L. prolificans* was studied. The signal detected using the pool of sera from mice infected with the infective dose was much higher than that of the contact dose. Likewise, the reactivity against the total extract was much higher than that obtained with the secretome (Appendix A). According to the results obtained, the dilution 1/1000 was selected for two-dimensional WB (2DWB), as it was high enough to detect reactivity in both groups and extracts.

The proteins of the secretome were separated by 2DE, where a total of 373 spots were detected, the largest number of them being localized in the p*I* between 3–7 (Figure 4A,C). After this, the pools of sera from the control group, the group inoculated with the contact dose and the one with the infective dose were studied against the secretome of *L. prolificans* by 2DWB (Figure 4B,D). Whereas no reactivity was observed with the control group (data not shown), three spots and thirty-six spots showed immunoreactivity against the contact dose and infective dose, respectively. No antigens shared by both groups were detected.

Subsequently, the three antigens detected with the sera from the contact dose, and the spots with a % volume higher than 2.5 recognized by infective dose sera were identified by LC-MS/MS (Table 1). In the case of spot 4, this was selected owing to its high reactivity against sera of mice infected with *Scedosporium* species, as explained below. Of these, twelve spots were successfully identified, which corresponded to seven different proteins: adenosylhomocysteinase (AdoHcyase) recognized by contact sera, and haloacid dehalogenase-like hydrolase (HAD-like hydrolase), Heat shock 70 kDa protein (Hsp70), nuclear movement protein nudC (NUDC), a glycosyl hydrolase belonging to the GH16 family (GH16), a cerato-platanin and a protein with unknown function (Hp jhhlp_006787) recognized by infected sera. Importantly, Hsp70 fragments with low *M*r were also identified, which according to the peptides detected, correspond to the final fragment of the protein. This fragment of the sequence showed a lower level of homology than the rest of the sequence with Hsp70s from human, *S. boydii, A. fumigatus* and *C. albicans* (Figure 5A).

### 3.4. Identification of the Immunodominant Antigens of the Total Extract of L. prolificans Recognized by Mice Inoculated with Contact and Infective Doses

Using the same methodology as above, the proteins of the *L. prolificans* total extract were separated by 2DE, where 944 spots were detected, with the greater number of spots being in the p*I* between 4–8 and *M*r of 25–130 kDa (Figure 4E). Referring to the antigen detection by 2DWB, two immunoreactive spots were recognized by the contact group sera and eighty-nine by the infective group sera (Figure 4F,H). No antigens shared by both groups were detected and the control group showed no recognition (data not shown).

Then, the two spots recognized by the contact dose sera and the seventeen spots with a % volume higher than 0.3 recognized by infective dose sera were identified by LC-MS/MS (Table 2). The nineteen immunoreactive spots corresponded to ten different proteins: two isoforms of proteosome subunit alpha as antigens associated with contact dose and dipeptidyl-peptidase (Dpp), Hsp70, proteasomal ubiquitin receptor, carboxypeptidase Y (CPY), RanBP1 domain-containing protein (RanBP1), proliferating cell nuclear antigen (PCNA), HAD-like hydrolase, vacuolar protein sorting-associated protein 28 (Vps28) and NFU1 iron-sulfur cluster scaffold-like protein (NFU1) as antigens associated with disseminated infection. Of all the antigens recognized by the infected sera according to % volume, Hsp70 was detected as the most immunodominant antigen with a value 20 times higher than the following antigen.

### 3.5. Study of the Localization, Function and Homology of L. prolificans Antigens

The bioinformatic analysis of the proteins detected in the secretome predicted the classical secretion of HAD-like hydrolase, GH16 and cerato-platanin (Figure 5C). The analyses also predicted adhesion properties for the GH16 and Hp jhhlp_006787 proteins. Among the proteins in the total extract, the classical secretion of CPY and Had-like hydrolase, and the mitochondrial localization of RanBP1 and NFU1 protein was predicted. Only the proteasomal ubiquitin receptor showed adhesion properties.

The study of the homology of *L. prolificans* proteins compared to those of *Scedosporium* spp., *A. fumigatus*, *C. albicans* and human proteins showed that the greatest similarities were obtained with *Scedosporium* spp, followed by *A. fumigatus*, obtaining fewer similarities with secreted antigens than with the total extract (Figure 5D). Among these, AdoHcyase and Hsp70 were the ones that showed the highest level of homology, obtaining in all cases identities higher than 70%. On the other hand, the proteins that showed a lower level of homology against the human ones were Vps28 (35%), proteasomal ubiquitin receptor (35%), CPY (32%), HAD-like hydrolase (32%), GH16 (31%), Hp jhhlp_006787 (0%) and cerato-platanin (0%). All mentioned antigens showed a homology <50% compared to *C. albicans*, and the last four showed a homology <50% compared to *A. fumigatus*.

According to the functionality, the antigens with non-defined function and the ones related to degradation processes (carbohydrate and amino acid metabolism) stand out in the secretome, whereas among the immunoreactive proteins identified in the total extract, the group containing antigens involved in protein degradation is the most abundant (Figure 5E).

### 3.6. Identification of the Antigens from the Secretome and Total Extract of L. prolificans Recognized by Sera from Mice Infected with S. boydii, S. aurantiacum and A. fumigatus and Study of the Prevalence of Hsp70 Recognition

To compare the humoral responses developed against *L. prolificans, S. boydii, S. aurantiacum* and *A. fumigatus*, the reactivity of sera from mice infected with each species was studied against its respective fungal extracts. In this way, a signal was detected up to 1/100,000 dilution with *L. prolificans, S. boydii* and *S. aurantiacum* and up to 1/10,000 with *A. fumigatus* (Appendix A). As a strong reactivity was observed in all cases, the dilution 1/1000 was selected for later studies by 2DWB.

The immunoreactive spots of both extracts, secretome and total extract, of *L. prolificans* recognized by sera from mice infected with *S. boydii* and *S. aurantiacum* showed the same pattern as when tested against sera from mice infected by *L. prolificans*, except for the higher reactivity obtained against the secreted Hsp70 (spot 1 in Figure 6), which corresponds to spot number 4 of the *L. prolificans* identifications (Figure 4 and Table 1). Hence, these antigens, were already identified in Section 3.3 and Section 3.4. In all 2DWB the Hsp70 was detected as the most immunodominant antigen. On the contrary, no recognition in the secretome and a very weak reaction in total extract of *L. prolificans* were detected using the sera from mice infected with *A. fumigatus*. In this case, none of the low-reactivity antigens coincided with the ones recognized using *Lomentospora/Scedosporium* group sera and according to previous immunoproteomic analyses carried out by our group it was demonstrated that enolase is one of them [11,14].

In order to study the humoral response against the most immunogenic antigen detected in more detail, the Hsp70, the reactivity of each individual mouse infected with *L. prolificans S. boydii, S. aurantiacum* and *A. fumigatus* against the purified antigen was studied by 1DWB (Figure 7). All the sera from mice infected with *L. prolificans, S. boydii* and *S. aurantiacum* clearly recognized the purified Hsp70. In the case of the control group (data not shown) and of mice infected with *A. fumigatus*, however, no recognition was detected.

## 4. Discussion

The infections caused by *L. prolificans* disseminated frequently (44.4%) and have high mortality rates (87.5%) [1], which is associated with the resistance of the fungus to currently available antifungals and the lack of rapid and reliable diagnostic techniques. Therefore, greater efforts must be made to identify new fungal molecules with medical applications that may result from immunoproteomic studies. With this in mind, the infection caused by *L. prolificans* in mice was compared with the one caused by the other fungal species *S. boydii, S. aurantiacum* and *A. fumigatus.* At the same time, the reactivity of the humoral responses in each case was studied against the extracts of *L. prolificans* so as to identify new therapeutic and diagnostic targets.

First, the murine model of infection by *L. prolificans* using between 10^2^ and 10^5^ conidia/animal showed that while in mice infected with low conidia doses the fungus was collected mainly in organs associated with blood circulation (spleen, liver, kidneys), with higher doses invasion of the brain and lungs was also observed. The high fungal load in the brain may be related to the neurotropism of *L. prolificans*, which tends to invade the Central Nervous System (CNS) and generally lead to death [1,15]. In fact, the group infected with the highest dose showed neurological abnormalities, such as leaning to one side, complete loss of balance and ataxia, similarly to those observed in previous trials [16,17]. This neurotropism has been related to an inefficient microglia response [18].

In turn, the infections caused by *S. boydii* and *S. aurantiacum* showed that compared to the 50% mortality rate of *L. prolificans, S. boydii* and *S. aurantiacum* caused a rate of 16.67%. This lower virulence of *Scedosporium* spp. when compared to *L. prolificans* has been described previously [19,20]. Turning to *A. fumigatus*, this species showed the lowest virulence and therefore, a higher dose was required to develop systemic infection, which caused 50% mortality. The organs in which the highest fungal load was detected were the kidneys, except for *S. aurantiacum.* Moreover, *L. prolificans* presented the greatest amount of conidia, probably due to its capacity for in vivo sporulation [1]. This ability is associated with the tendency of infections to disseminate and may explain the higher fungal load in the brain, which could be also favored by a lower capacity of the microglia to phagocyte *L. prolificans*, compared to *S. boydii* and *S. aurantiacum* [18].

Subsequently, the analysis of the humoral responses developed against a contact dose (10^2^ conidia/animal) and against an infective dose (10^5^ conidia/animal) of *L. prolificans* was carried out by 2DWB. The sera of both groups were first studied against the secretome of *L. prolificans*, which is very important because fungal secreted proteins have a role in virulence and little homology with human proteins [21], and may contain specific diagnostic markers, therapeutic targets or vaccine candidates. In the proteomic analysis, the twelve most immunoreactive spots were identified, these corresponded to seven different antigens. Specifically, adenosylhomocysteinase (AdoHcyase) was identified in the contact dose, and haloacid dehalogenase-like hydrolase (HAD-like hydrolase), Heat shock 70 kDa protein (Hsp70), nuclear movement protein (NUDC), a glycosyl hydrolase belonging to the GH16 family (GH16), a cerato-platanin and a protein with unknown function (Hp jhhlp_006787) in the infective dose. The identification of proteins with unknown function in the secretome of filamentous fungi is common and moreover, in the case of this protein is logical taking into account its low-level of homology with other fungal species such as *A. fumigatus* (38.22%).

The antigen recognized by contact sera, AdoHcyase, is a metabolic enzyme responsible for the reversible hydration of S-adenosyl-L-homocysteine in adenosine and homocysteine [22] and has been detected as exoproteome antigen in other fungal species [23]. This antigen could probably be also recognized by infective dose sera, but the high immunoreactivity of other proteins might impede its detection. The fact that immunodominant antigens are not shared by both groups raises the significance of those recognized by mice with disseminated infection as infection markers.

Bioinformatic analysis of the identified proteins showed that secretion by the classical route was only predictable in three of them (42.9%): HAD-like hydrolase, GH16 and cerato-platanin. Numerous studies have detected extracellular proteins without a signal peptide that could be secreted by alternative routes, by exocytosis of coated vesicles, secretory lysosomes, microvesicles or ATP-binding cassette transporters [24] which could explain the presence of the rest of the proteins. In fact, the secretion of AdoHcyase and Hsp70 by extracellular vesicles has already been described for several pathogenic fungi [25,26]. On the other hand, two proteins, GH16 and Hp jhhlp_006787 were found to have adhesion properties, which in extracellular proteins has been related to facilitate fungus-host interaction and host invasion [27,28].

In relation to their function, 50% of those with known functions are hydrolytic enzymes. Specifically, two hydrolases, AdoHcyase and HAD-like hydrolase, and one glycosylase were identified. The HAD-like hydrolase superfamily includes dehalogenases, phosphonatases, phosphomutases, phosphatases and ATPases [29]. Some of them have been detected in the secretome of *S. boydii* [12], identified as seroprevalent antigens of *L. prolificans* [14] and related to morphology, cell wall integrity, thermotolerance, adhesion and virulence of fungal pathogens [30,31].

A protein belonging to the cerato-platanin family was also detected, formed by small, secreted and rich in cysteine proteins exclusively synthesized by filamentous fungi [32], some of them pathogens [12,23,33], and described as necessary for inflammation and the remodelling of the airways caused by *A. fumigatus* [34].

Likewise, the NUDC, which has been previously located in the cortex of hyphae and at spindle pole bodies, has been identified as a protein necessary for nuclear movement and hyphal growth [35]. Finally, Hsp70 is one of the most important antigens of the secretome of *L. prolificans*. However, taking into account its immunoreactivity also in the total extract of *L. prolificans* too and in the interspecies comparison described in the following paragraphs, it will be discussed in detail below.

After the immunoproteomic analysis of the secretome, the same process was carried out with the total extract of *L. prolificans*. In the analysis of the proteome, the nineteen immunoreactive spots with the highest % volume were identified and corresponded to ten different proteins: two isoforms of the proteosome subunit alpha as antigens associated with contact with the fungus, and dipeptidyl-peptidase (Dpp), Hsp70, proteasomal ubiquitin receptor, carboxypeptidase Y (CPY), RanBP1 domain-containing protein (RanBP1), proliferating cell nuclear antigen (PCNA), HAD-like hydrolase, vacuolar protein sorting-associated protein 28 (Vps28) and NFU1 iron-sulfur cluster scaffold-like protein (NFU1) as antigens associated with disseminated infection.

Of all the antigens recognized by the infected sera, according to % volume, Hsp70 was detected as the most immunodominant antigen with a value 20 times higher than the second-placed antigen. This protein and HAD-like hydrolase have also been identified in the secretome recognized by sera from infected mice, and together with Dpps have already been identified as antigens of fungi [14,33,36,37,38]. Furthermore, Dpps have been selected as an interesting targets for serodiagnosis of different aspergillosis such as allergic bronchopulmonary aspergilosis (ABPA), invasive *A. terreus* infection and aspergilloma [21].

Regarding the other antigens identified in the total extract, the Vps28 is part of the endosomal sorting complex required for transport (ESCRT-I) that is associated with endosomal membranes and participates in sorting proteins from endosomes to vacuole through multivesicular bodies [39]. The deletion of Vps28 protein in *C. albicans* increases its susceptibility to echinocandins and azoles [40], and significantly reduces its virulence in a murine model of invasive candidiasis [41]. Furthermore, as far as we know, it is the first time that RanBP1, PCNA and NFU1 have been detected as fungal antigens. In terms of functionality, RanBP1 which is involved in transport between the nucleus and the cytoplasm, is essential for cell viability and required for both nuclear protein import and poly(A)+ RNA export in fungi [42]. On the other hand, PCNA is a conserved protein that serves as a DNA docking platform for many proteins that function in DNA replication, repair, recombination, cell cycle, and chromatin remodelling [43]. NFU1 is located in the mitochondria and plays an important role in intracellular iron homeostasis and Fe/S protein biosynthesis [44,45].

Finally, to study the specificity of *L. prolificans* antigens, sera from mice infected with *S. boydii, S. aurantiacum* and *A. fumigatus* were used against the total extract and secretome of *L. prolificans*. The results showed that sera obtained from mice infected with species of *Scedosporium* spp. recognized almost exactly the same immunoreactive proteins as those infected with *L. prolificans*. All these proteins have already been described earlier in this discussion. This high cross-reactivity makes it impossible to find species-specific antigens inside the *Scedosporium/Lomentospora* group and this is most likely due to the high level of homology associated with the phylogenetic proximity between them. In contrast, sera from mice infected with *A. fumigatus* reacted with very few proteins of *L. prolificans*, including enolase. This result is not surprising, since this antigenic protein is highly conserved, showing cross-reactivity between several fungal species [11,46,47].

The antigens shared between the species *S. boydii, S. aurantiacum* and *L. prolificans* are interesting candidates for the study of alternative therapies for these infections, as well as potential diagnostic markers for differentiation from *A. fumigatus*. To be specific, from the total extract of *L. prolificans*, the proteasomal ubiquitin receptor, CPY, Vps28 and HAD-like hydrolase, present a lower than 40% homology with human proteins, which underlines their interest as therapeutic targets or to produce vaccines. On the other hand, among those secreted, the protein of unknown function, the GH16 and the cerato-platanin have low or no homology with human proteins, and less than 50% with those of *A. fumigatus*. Therefore, in addition to therapeutic targets, they are also interesting for the development of specific diagnostic techniques for *Lomentospora/Scedosporium*. Since they are secreted proteins, the direct detection of these proteins in body fluids could be of interest since many of the patients affected by these mycoses present a defective immune response.

Hsp70 deserves a special mention, as it was described as the most immunodominant antigen of the total extract of *L. prolificans*, and has also been recognized by mice infected with *Scedosporium/Lomentospora* species but not with *A. fumigatus*. The presence of Hsp70 on the cell surface [14] and its secretion makes it very accessible to the immune response and could explain its high immunogenicity. In addition, proteins of this family have been previously described as antigens [36,37,38] or virulence factors of important pathogenic fungi [48,49].

It is worth highlighting the interest in using Hsp70 as a diagnostic target for *Lomentospora/Scedosporium* since it is the most immunodominant antigen of *L. prolificans*. Therefore, the Hsp90 has already been used for the development of diagnostic methods for invasive *C. albicans* infections [50,51].

In addition, Hsp70 has been identified as a prevalent antigen of *L. prolificans* recognised by healthy humans [11,14], which increases its interest for the development of vaccines. In fact, Hsps have already been studied as fungal vaccines [52,53] and even used as adjuvants or carrier molecules in the development of vaccines against bacteria, virus and parasites [54,55].

Although Hsp70 is a highly conserved protein, as observed in the homology study, it presents a C- terminal fragment of lower similarity. Studies carried out with the Hsp70 of *Mycobacterium tuberculosis* showed that this protein stimulates the cells of the immune response, inducing the production of chemokines and cytokines in human monocytes, and the maturation of dendritic cells, and that the C-terminal domain is responsible for such stimulation [56]. Therefore, the study of the final fragment of this protein as a vaccine is of special interest, due to its potential to induce the immune response and its lower level of homology with respect to human protein, which could decrease the appearance of side effects associated with cross-reactivity

Finally, the Hsp70 of *L. prolificans* has been also related to the resistance to diverse antifungals [57,58,59] and therefore, could be useful for the development of alternative treatments. With this in mind, the monoclonal antibody Mycograb, produced against *Candida* Hsp90 in combination with amphotericin B has demonstrated efficacy against *C. albicans* and *C. neoformans* [60,61]. In addition, inhibitory compounds of the Hsp90-calcineurine axis, which includes Hsp70, have been successfully tested in vitro against resistant strains of *A. fumigatus* [62] and *L. prolificans* [63].

## 5. Conclusions

In conclusion, *L. prolificans* was the most virulent species in this study, followed by *Scedosporium* spp. and finally by *A. fumigatus*. Infections caused by *L. prolificans* were characterized by a greater presence of conidia in vivo and a tendency to infect the brain. All antigens recognized by sera from mice infected with *L. prolificans* showed cross-reactivity with sera from mice infected with *Scedosporium* spp., which makes differential diagnosis between these species difficult using serological techniques but makes it possible to differentiate them from *A. fumigatus*. Among the shared antigens, proteasomal ubiquitin receptor, CPY, Vps28, HAD-like hydrolase, GH16, Hp jhhlp_006787 and cerato-platanin are interesting candidates for study not only for diagnosis, but also as therapeutic targets or even for the development of vaccines due to their null or low homology with human proteins. Another protein that deserves special mention is Hsp70, recognized by all mice infected by *L. prolificans* and *Scedosporium* spp. Despite being a highly conserved protein, it presents a C-terminal fragment that is less similar and whose clinical potential should be studied in future research.

## Figures and Tables

**Figure 1 vaccines-07-00212-f001:**
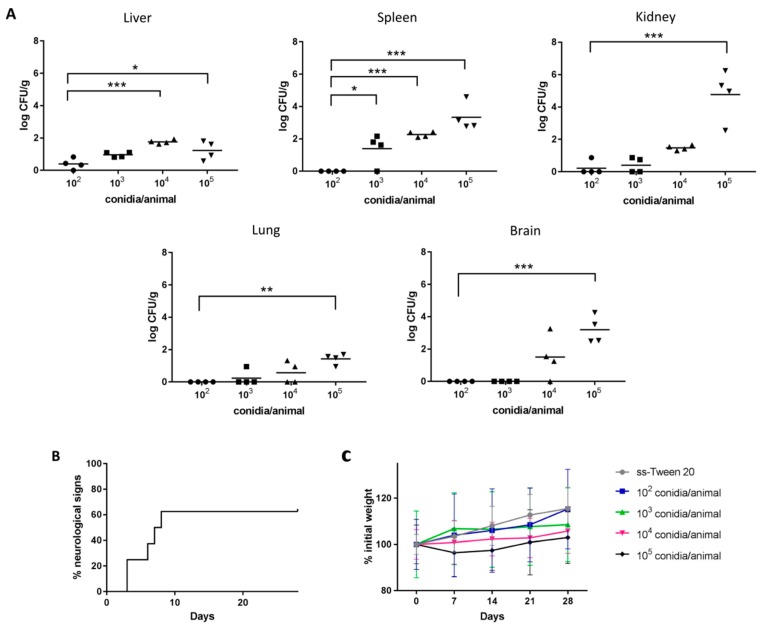
Fungal tissue burden results (**A**) and animal welfare monitoring (**B**,**C**) after intravenous infection with *L. prolificans* 10^2^–10^5^ conidia/animal. Neurological signs detection in mice infected with 10^5^ conidia/animal (**B**). Weekly weighing of mice inoculated with *L. prolificans* and of control group (**C**). The fungal tissue burdens were compared using the two-way ANOVA test, taking as reference the group infected with 10^2^ conidia/animal; * *p* < 0.05, ** *p* < 0.01 and *** *p* < 0.001.

**Figure 2 vaccines-07-00212-f002:**
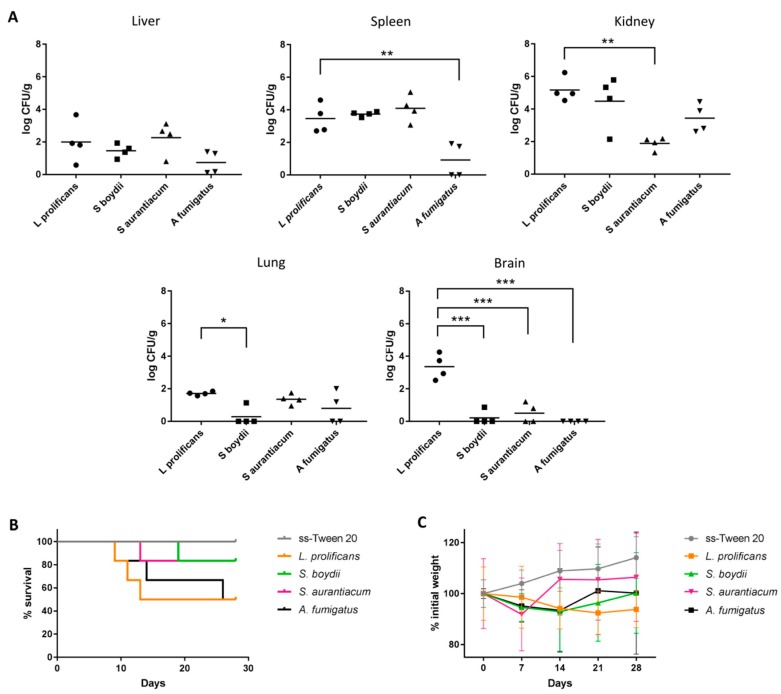
Fungal tissue burden results (**A**), survival (**B**) and weekly weighing (**C**) of mice intravenously infected with 10^5^ conidia/animal of *L. prolificans, S. boydii* and *S. aurantiacum,* and with *A. fumigatus* 5 × 10^6^ conidia/animal. The fungal tissue burdens were compared against *L. prolificans* group using the two-way ANOVA test; * *p* < 0.05, ** *p* < 0.01 and *** *p* < 0.001.

**Figure 3 vaccines-07-00212-f003:**
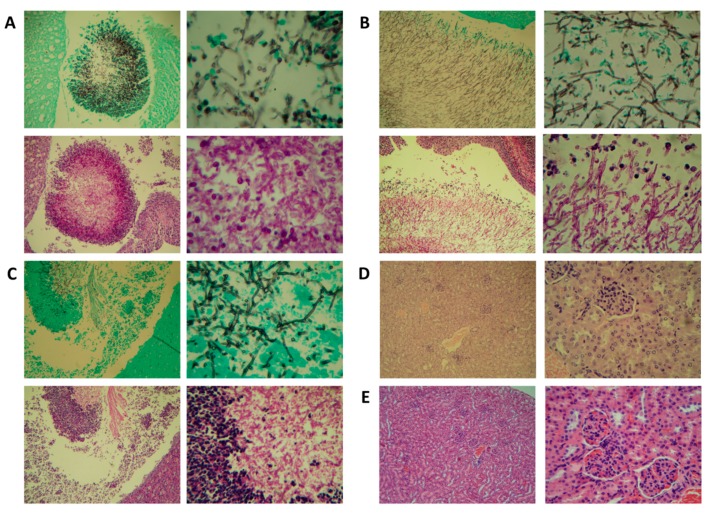
Representative histopathological Grocott methenamine silver (above) and hematoxylin-eosin (bellow) stained sections of kidneys from mice intravenously infected with 10^5^ conidia/animal of *L. prolificans* (**A**), *S. boydii* (**B**) and *S. aurantiacum* (**C**). Representative histopathological hematoxylin-eosin stained sections of kidneys from *A. fumigatus* 5 × 10^6^ conidia/animal infected group (**D**) and from control group (**E**).

**Figure 4 vaccines-07-00212-f004:**
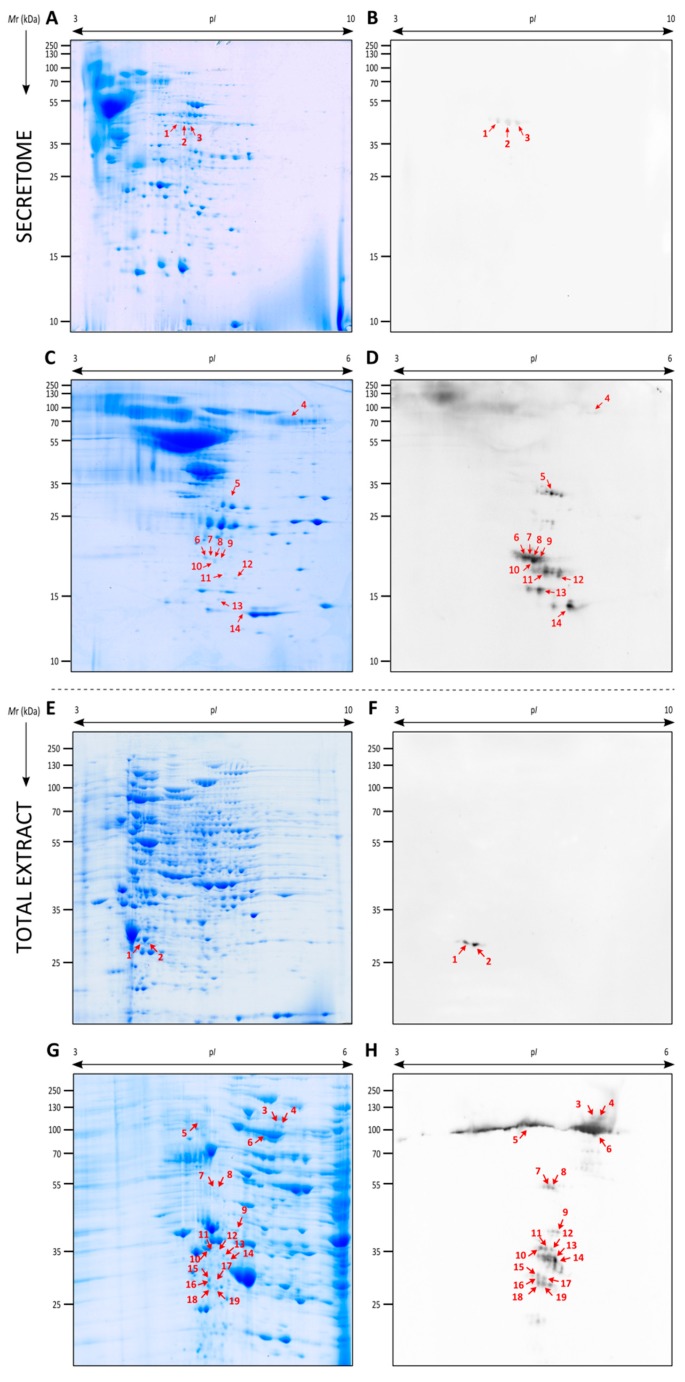
Representative two-dimensional electrophoresis (2DE) proteome and immunome of the secretome (**A**–**D**) and total extract (**E**–**H**) from *L. prolificans.* Secreted proteins resolved by 2DE and Coomassie Brilliant Blue (CBB) stained (**A**,**C**) or electrotransferred to PVDF membranes to detect immunoreactive proteins recognized by mice inoculated with the contact (**B**) and infective dose (**D**). Total extract proteins resolved by 2DE and CBB stained (**E**,**G**) or electrotransferred to PVDF membranes to detect immunoreactive proteins recognized by mice inoculated with the contact (**F**) and infective dose (**H**).

**Figure 5 vaccines-07-00212-f005:**
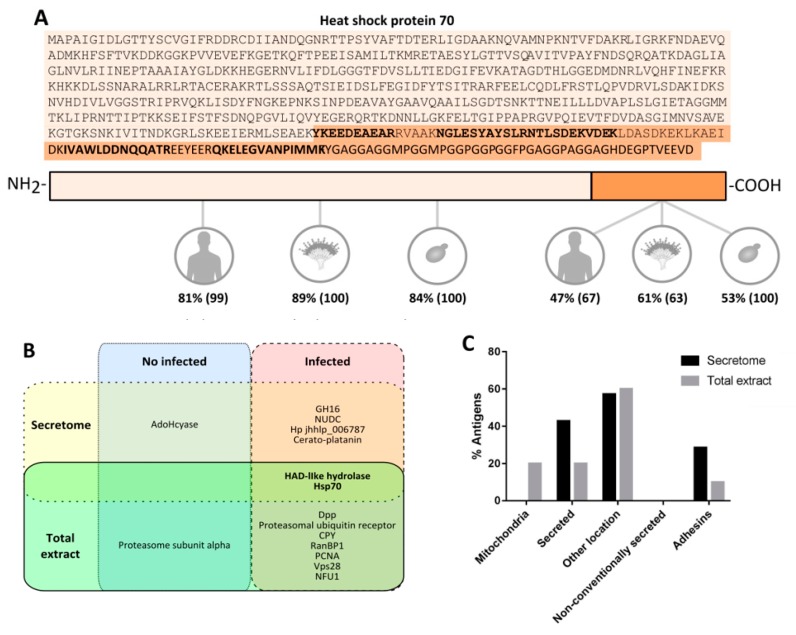
Analysis of secretome and total extract antigens of *L. prolificans.* Location of the low molecular weight secreted fragments of Hsp70 in the protein sequence, according to the bold marked peptides identified by LC-MS/MS. Study of the similarity of the Hsp70 C-terminal and of the remaining fragment against the corresponding human, *A. fumigatus* and *C. albicans* Hsp70s. Percentages of identities and covertures (in brackets) are indicated (**A**). Diagram of *L. prolificans* secretome and total extract antigens that reacted with sera of mice inoculated with contact and infective dose (**B**). Bioinformatic analysis of their location (**C**) and of their similarity compared to human, *S. boydii, A. fumigatus* and *C. albicans* proteins, indicating identity percentages (**D**). Functional classification of the secretome (above) and total extract (bellow) antigens (**E**).

**Figure 6 vaccines-07-00212-f006:**
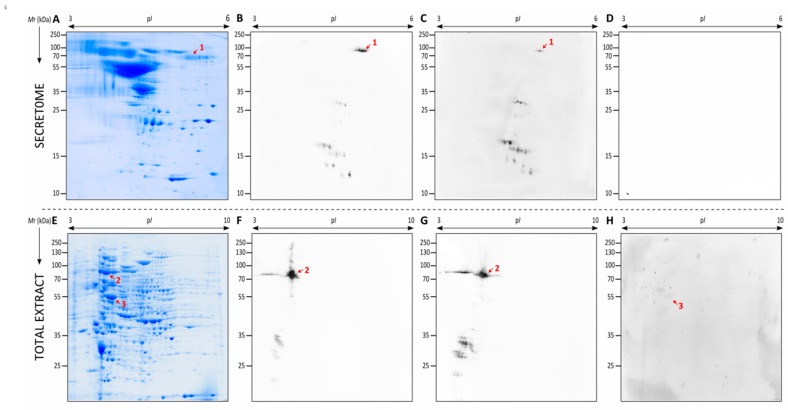
Cross-reactivity study of the proteins of *L. prolificans* recognized by sera of mice infected with *Scedosporium boydii*, *Scedosporium aurantiacum* and *Aspergillus fumigatus.* Representative proteome of secretome from *L. prolificans* (**A**) and the detection *L. prolificans* proteins recognized by mice infected with 10^5^ conidia/animal of *S. boydii* (**B**) and *S. aurantiacum* (**C**), and with *A. fumigatus* 5 × 10^6^ conidia/animal (**D**). Representative proteome of total extract from *L. prolificans* (**E**) and detection of *L. prolificans* proteins recognized by mice infected with 10^5^ conidia/animal of *S. boydii* (**F**) and *S. aurantiacum* (**G**), and with *A. fumigatus* 5 × 10^6^ conidia/animal (**H**). Hsp70 from the secretome (1), and Hsp70 (2) and enolase (3) from the total extract are marked.

**Figure 7 vaccines-07-00212-f007:**
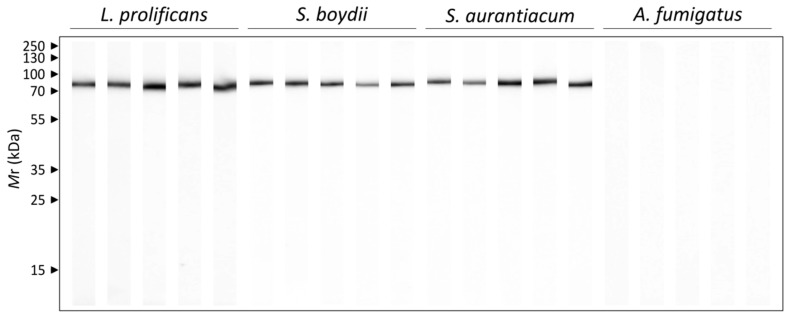
Humoral response of mice infected with 10^5^ conidia/animal of *L. prolificans*, *S. boydii* and *S. aurantiacum*, and with *A. fumigatus* 5 × 10^6^ conidia/animal against the purified Hsp70 of *L prolificans*. Five sera per group were used individually.

**Table 1 vaccines-07-00212-t001:** Identification by LC-MS/MS of immunodominant antigens from *Lomentospora prolificans* secretome reacting with serum IgG of mice inoculated with the contact and infective dose of this fungus.

Spot No.	NCBI No.	Protein Name	Microorganism	Matching Peptides	Sequence Coverage (%)	MASCOT Score	Theor. p*I*/*M*r (kDa)	Exper. p*I/M*r (kDa)
**Secreted antigens of *L. prolificans* reacting with serum of mice inoculated with the contact dose (Figure 4A)**								
**1**		-						5.62/46.98
**2**	PKS06692.1	Adenosylhomocysteinase (hp jhhlp_006766)	*L. prolificans*	1	5	156	5.77/49.09	5.79/46.61
**3**		-						5.90/46.24
**Secreted antigens of *L. prolificans* reacting with serum of mice inoculated with the infective dose (Figure 4C)**								
**4**	PKS06780.1	Heat shock 70 kDa protein (hp jhhlp_006854)	*L. prolificans*	27	51	1758	5.11/71.50	5.16/76.79
**5**	PKS07974.1	Haloacid dehalogenase-like Hydrolase (hp jhhlp_006586)	*L. prolificans*	4	35	179	4.93/26.36	4.57/30.28
**6**	PKS06780.1	Heat shock 70 kDa protein (hp jhhlp_006854)	*L. prolificans*	5	7	170	5.11/71.50	4.25/18.45
**7**	PKS06780.1	Heat shock 70 kDa protein (hp jhhlp_006854)	*L. prolificans*	5	8	160	5.11/71.50	4.29/18.45
**8**	PKS06780.1	Heat shock 70 kDa protein (hp jhhlp_006854)	*L. prolificans*	5	8	172	5.11/71.50	4.36/18.41
**9**	PKS07141.1	Glycosyl hydrolase family 16 protein (Hp jhhlp_005741)	*L. prolificans*	2	7	90	4.44/41.13	4.43/18.43
	AAC49800.1	Glyceraldehyde-3-phosphate dehydrogenase	*C. albicans*	1	8	59	6.33/35.51	
**10**	PKS06780.1	Heat shock 70 kDa protein (hp jhhlp_006854)	*L. prolificans*	1	1	54	5.11/71.50	4.35/17.95
**11**	PKS06780.1	Heat shock 70 kDa protein (hp jhhlp_006854)	*L. prolificans*	3	5	131	5.11/71.50	4.50/16.83
	PKS12714.1	Nuclear movement protein nudC (hp jhhlp_000922)	*L. prolificans*	1	5	81	5.30/21.12	
**12**	PKS12714.1	Nuclear movement protein nudC (hp jhhlp_000922)	*L. prolificans*	2	6	108	5.30/21.12	4.82/16.18
**13**	PKS06713.1	Hp jhhlp_006787	*L. prolificans*	1	7	107	4.82/22.77	4.42/14.90
**14**	PKS07725.1	Cerato-platanin (Hp jhhlp_006333)	*L. prolificans*	2	23	256	5.37/14.03	4.79/13.45

**Table 2 vaccines-07-00212-t002:** Identification by LC-MS/MS of immunodominant antigens from *Lomentospora prolificans* total extract reacting with serum IgG of mice inoculated with the contact and infective dose of this fungus.

Spot No.	NCBI No.	Protein Name	Microorganism	Matching Peptides	Sequence Coverage (%)	MASCOT Score	Theor. p*I/M*r (kDa)	Exper. p*I*/*M*r (kDa)
**Antigens of *L. prolificans* total extract reacting with serum of mice inoculated with the contact dose (Figure 4E)**								
**1**	PKS10679.1	Proteasome subunit alpha (hp jhhlp_002435)	*L. prolificans*	5	22	304	5.28/28.04	4.73/28.35
	PKS08308.1	NADP oxidoreductase coenzyme F420-dependent (hp jhhlp_005252)	*L. prolificans*	5	27	288	5.66/32.60	
**2**	PKS10679.1	Proteasome subunit alpha (hp jhhlp_002435)	*L. prolificans*	10	41	474	5.28/28.04	4.91/28.32
**Antigens of *L. prolificans* total extract reacting with serum of mice inoculated with the infective dose (Figure 4G)**								
**3**	PKS06544.1	Dipeptidyl-peptidase (hp jhhlp_007292)	*L. prolificans*	13	20	602	5.19/79.99	5.26/81.64
**4**	PKS06544.1	Dipeptidyl-peptidase (hp jhhlp_007292)	*L. prolificans*	15	23	797	5.19/79.99	5.32/81.10
**5**	PKS06780.1	Heat shock 70 kDa protein (hp jhhlp_006854)	*L. prolificans*	22	49	1658	5.11/71.50	4.08/83.83
**6**	PKS06780.1	Heat shock 70 kDa protein (hp jhhlp_006854)	*L. prolificans*	23	43	1159	5.11/71.50	4.66/77.14
**7**	PKS10466.1	Proteasomal ubiquitin receptor (hp jhhlp_002217)	*L. prolificans*	5	25	225	4.65/40.76	4.62/51.74
**8**	PKS12946.1	Carboxypeptidase (hp jhhlp_000287)	*L. prolificans*	2	4	111	4.99/60.56	4.66/51.68
**9**	PKS13321.1	RAN-specific GTPase-activating protein (hp jhhlp_000092)	*L. prolificans*	9	40	607	5.11/28.07	4.76/35.25
	PKS12026.1	60S acidic ribosomal protein P0 (hp jhhlp_001322)	*L. prolificans*	4	20	374	4.81/33.75	
**10**	PKS10574.1	Proliferating cell nuclear antigen (hp jhhlp_002328)	*L. prolificans*	9	39	449	4.56/28.92	4.36/32.09
**11**	PKS10574.1	Proliferating cell nuclear antigen (hp jhhlp_002328)	*L. prolificans*	3	18	326	4.56/28.92	4.41/32.01
**12**	PKS10574.1	Proliferating cell nuclear antigen (hp jhhlp_002328)	*L. prolificans*	2	11	118	4.56/28.92	4.45/32.18
	XP_016642844.1	NAC domain-containing protein (hp SAPIO_CDS5122)	*S. apiospermum*	2	13	100	4.59/22.31	
**13**	PKS07974.1	Haloacid dehalogenase-like hydrolase (hp jhhlp_006586)	*L. prolificans*	5	31	390	4.93/26.36	4.51/30.62
**14**	PKS07974.1	Haloacid dehalogenase-like hydrolase (hp jhhlp_006586)	*L. prolificans*	7	41	431	4.93/26.36	4.41/30.00
**15**	PKS10466.1	Proteasomal ubiquitin receptor (hp jhhlp_002217)	*L. prolificans*	3	12	174	4.65/40.76	4.57/30.19
	PKS11928.1	Eukaryotic translation initiation factor 6 (hp jhhlp_001223)	*L. prolificans*	4	25	154	4.33/23.13	
**16**	PKS10466.1	Proteasomal ubiquitin receptor (hp jhhlp_002217)	*L. prolificans*	4	12	241	4.65/40.76	4.57/29.58
	PKS11928.1	Eukaryotic translation initiation factor 6 (hp jhhlp_001223)	*L. prolificans*	4	26	214	4.33/23.13	
**17**	PKS12778.1	Vacuolar protein sorting-associated protein 28 (hp jhhlp_000989)	*L. prolificans*	4	19	157	4.68/27.98	4.64/29.58
	PKS09814.1	Aminopeptidase (hp jhhlp_004437)	*L. prolificans*	3	7	129	5.50/60.03	
**18**	PKS09755.1	NFU1 iron-sulfur cluster scaffold-like protein (hp jhhlp_004376)	*L. prolificans*	5	28	299	5.78/33.19	4.61/24.42
**19**	PKS09755.1	NFU1 iron-sulfur cluster scaffold-like protein (hp jhhlp_004376)	*L. prolificans*	6	31	343	5.78/33.19	4.64/28.20

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
