# Peer review of "Study of Humoral Responses against Lomentospora/Scedosporium spp. and Aspergillus fumigatus to Identify L. prolificans Antigens of Interest for Diagnosis and Treatment"

_vaccines, 2019, doi:10.3390/vaccines7040212_

Round 1

Reviewer 1 Report

The authors investigated for potential vaccine candidates from L. prolificans antigens which is an important and interesting topic. The study seems to be robustly performed and discussions are well supporting the results. Authors are suggested to proofread the manuscript once more by a Native English speaker for correcting minor grammatical errors that are presented in the manuscript.

Author Response

Response to Reviewer 1 Comments

Point 1: The authors investigated for potential vaccine candidates from L. prolificans antigens which is an important and interesting topic. The study seems to be robustly performed and discussions are well supporting the results. Authors are suggested to proofread the manuscript once more by a Native English speaker for correcting minor grammatical errors that are presented in the manuscript.

The manuscript has been checked again by a native English speaking professional.

Reviewer 2 Report

In the manuscript by Buldain et al., the authors developed a disseminated murine infection model to compare the virulence of L. prolificans with that of Scedosporium boydii, Scedosporium aurantiacum and Aspergillus fumigatus. As a major result of the study, the authors identified the antigens to be studied as therapeutic and diagnostic targets. Overall the manuscript is well written, the experimental procedures and methods are fully described, the performed experiments are rigorous, I also agree with all conclusions made by the authors. However, I have a few comments that have to be addressed before this manuscript.

I would strongly suggest running statistics on the data presented in Figure 1 and Figure 2. Figure 3 shows representative images of kidney sections, however, statics for this data is not reported. Please indicate n for each sample. Figure 3 shows representative images of the kidney for infected animals; however, no control tissue images are presented. Figure 3 shows only images of the kidney, however, the authors collected other organs for analysis. I would recommend presenting histological analysis for other collected issues for the infected and non-infected animals. For all studies, the authors used 8-week-old female mice. Please provide the rationale for the gender and strain used, as well as age. Please make a statement if the experimenter was blinded to the condition of each mice group used in the studies.

Author Response

Response to Reviewer 2 Comments

In the manuscript by Buldain et al., the authors developed a disseminated murine infection model to compare the virulence of L. prolificans with that of Scedosporium boydii, Scedosporium aurantiacum and Aspergillus fumigatus. As a major result of the study, the authors identified the antigens to be studied as therapeutic and diagnostic targets. Overall the manuscript is well written, the experimental procedures and methods are fully described, the performed experiments are rigorous, I also agree with all conclusions made by the authors. However, I have a few comments that have to be addressed before this manuscript.

I would strongly suggest running statistics on the data presented in Figure 1 and Figure 2.

We agree with the reviewer in that statistics in these figures is very informative and, therefore, we have included it.

Figure 3 shows representative images of kidney sections, however, statics for this data is not reported. Please indicate n for each sample.

The paragraph regarding this comment has been modified in the Material an Methods section as following: “To perform the histological study, the organs of all the animals were fixed in 10% formalin and immersed in paraffin. Then, at least five different cuts, four micrometers wide, were stained with hematoxylin-eosin and Grocott’s methenamine silver.”

Figure 3 shows representative images of the kidney for infected animals; however, no control tissue images are presented.

According to reviewer´s comment, we have added images of the kidneys from the control group stained with hematoxylin-eosin.

Figure 3 shows only images of the kidney, however, the authors collected other organs for analysis. I would recommend presenting histological analysis for other collected issues for the infected and non-infected animals.

It is common to detect elevated CFU counts in the organs, but not to detect the infection by histological study, since the latter method has a lower sensitivity. In our study, histological sections showed only renal infection, and therefore, the other organs are not shown. We do not consider interesting for the article to add the other four images that will be very similar to those of the control group.

For all studies, the authors used 8-week-old female mice. Please provide the rationale for the gender and strain used, as well as age.

Immunocompentent female mice of 6-8 weeks are usually used in disseminated Scedosporium/Lomentospora infection models [1,2]. Moreover, the female mice are less aggressive and easier to handle.

Harun et al. Scedosporium aurantiacum is as virulent as S. prolificans, and shows strain-specific virulence differences, in a mouse model. Med. Mycol. 2010, 48 Suppl 1, S45–S51. Simitsopoulou et al. Antifungal activities of posaconazole and granulocyte-macrophage colony-stimulating factor ex vivo and in mice with disseminated infection due to Scedosporium prolificans. Antimicrob. Agents Chemother. 2004, 48, 3801–3805.

We selected Swiss mice because it is an outbred model and it has higher genetic variability than inbred mice. This greater variability allows us to obtain a more complete picture of the humoral response.

Please make a statement if the experimenter was blinded to the condition of each mice group used in the studies.

Animal welfare monitoring was always carried out by two researchers: one of them was the researcher who carried out the infections and, therefore, knew the condition of each group of mice, but the other one was blinded. In this way, the possible subjectivity was limited.

Furthermore, the researcher who performed the histological analysis was blinded to the condition of each group of mice used in the studies.